# NRVC: Neural Representation for Video Compression with Implicit Multiscale Fusion Network

**DOI:** 10.3390/e25081167

**Published:** 2023-08-04

**Authors:** Shangdong Liu, Puming Cao, Yujian Feng, Yimu Ji, Jiayuan Chen, Xuedong Xie, Longji Wu

**Affiliations:** School of Computer Science, Nanjing University of Posts and Telecommunications, Nanjing 210023, China; lsd@njupt.edu.cn (S.L.); 1021041408@njupt.edu.cn (P.C.); 2020070134@njupt.edu.cn (Y.F.); 1220044922@njupt.edu.cn (J.C.); 1221045716@njupt.edu.cn (X.X.); 1221045619@njupt.edu.cn (L.W.)

**Keywords:** video compression, implicit neural representation, attention mechanism

## Abstract

Recently, end-to-end deep models for video compression have made steady advancements. However, this resulted in a lengthy and complex pipeline containing numerous redundant parameters. The video compression approaches based on implicit neural representation (INR) allow videos to be directly represented as a function approximated by a neural network, resulting in a more lightweight model, whereas the singularity of the feature extraction pipeline limits the network’s ability to fit the mapping function for video frames. Hence, we propose a neural representation approach for video compression with an implicit multiscale fusion network (NRVC), utilizing normalized residual networks to improve the effectiveness of INR in fitting the target function. We propose the multiscale representations for video compression (MSRVC) network, which effectively extracts features from the input video sequence to enhance the degree of overfitting in the mapping function. Additionally, we propose the feature extraction channel attention (FECA) block to capture interaction information between different feature extraction channels, further improving the effectiveness of feature extraction. The results show that compared to the NeRV method with similar bits per pixel (BPP), NRVC has a 2.16% increase in the decoded peak signal-to-noise ratio (PSNR). Moreover, NRVC outperforms the conventional HEVC in terms of PSNR.

## 1. Introduction

In recent years, due to the powerful feature learning ability of deep networks and the robustness defects of manually designed traditional video compression technology modules [1,2,3], more and more researchers are no longer satisfied with changing a single module and have begun to focus on building an end-to-end video compression deep model [4]. By implementing all modules with deep neural networks and directly using end-to-end optimization rate-distortion objective functions, the global optimal solution can be obtained more quickly. However, these end-to-end deep compression methods [5,6,7] remain within the framework of conventional video compression methods, only replacing the manually designed modules with deep models for global joint optimization. Due to the utilization of optical flow estimation for obtaining sufficient contextual motion information to reconstruct the current frames [7], the entire compression pipeline becomes long and complex, leading to a large number of redundant parameters.

As a novel video compression approach based on implicit neural representation (INR), NeRV [8] transforms optimizes conventional video compression frameworks and simplifies compression pipelines. INR, as a novel and effective neural representation method, can encode a variety of input information to a function, which maps the input to the desired output and parameterizes the function with a deep neural network [8,9]. The advantage of INR is that it provides a genuine continuous version of the input signal, as opposed to its cropped and quantized counterpart [10]. Compared with traditional pixel-wise INR methods [10,11], NeRV is capable of performing image-wise processing on input videos. NeRV eliminates the need for inter-frame and intra-frame encoding, bypassing the redundant steps of reconstructing interval frames through keyframes. As a result, it shortens the end-to-end pipeline while maintaining the ability to reconstruct high-resolution and high-fidelity video frame images.

However, the NeRV network is limited by its single-scale feature pipeline and reliance solely on the MLP network for feature extraction. These restrictions hamper NeRV from accurately fitting the input, making it inadequate for video frame reconstruction and representation. Meanwhile, those also make it challenging for NeRV to handle more complex video frame data. Therefore, we propose a new model of a neural representation approach for the video compression with implicit multiscale fusion network (NRVC) which effectively utilizes features of different scales in the model for video frame representation. By fitting each video frame index on the video timeline with the network model, NRVC allows us to obtain of a highly effective function for representing input videos. The decoder can output any RGB video frame by a simple feedforward operation.

The main difference between the proposed NRVC and NeRV networks lies in the structure after the MLP layer. In NeRV, after completing feature extraction, five NeRV blocks are added in sequence, performing simple upsampling operations to restore the original video resolution and complete the video frame reconstruction. In contrast, after the MLP layer, NRVC achieves not only the goal of restoring the original video frame resolution, but also utilizes the MSRVC network, constructed with the channel attention mechanism and the multi-scale normalization network, to achieve more refined feature extraction. This improvement leads to enhanced accuracy in the video frame reconstruction process.The proposed MSRVC network contains two main blocks: normalized residual for video compression (NoRVC) and feature extraction channel attention (FECA).

Firstly, utilizing the normalized network, the proposed NoRVC model incorporates feature inputs from other layers during video frame reconstruction, thereby expanding the information content of the features. Normalization networks have shown outstanding capabilities in extracting multi-level style features in style transfer research [12], allowing for the creation of high-quality stylized images. The implementation of multiple NoRVC blocks aids in enhancing the network’s capability to reconstruct video frames in MSRVC, effectively addressing the limited feature expression problem observed in NeRV.

Secondly, we utilize the FECA attention mechanism to enhance the attention of feature channels and capture interaction information among feature extraction channels. Moreover, in order to improve the effectiveness of the final feature information, we weigh the corresponding feature map channels based on their effectiveness [13,14].

In NRVC, all video frame images are transformed into 80-dimensional normalized position vectors with the position encoding layer, which serve as inputs to the MLP. The hidden layer of the MLP has a size of 512, and the output layer’s feature size depends on the target video resolution, with common sizes being 16 × 9 and 4 × 3, and the number of channels is 48. The entire MSRVC network consists of one FECA block and five NoRVC blocks. After the FECA module adaptively extracts features, the outputs are used as inputs to the first NoRVC layer and as additional features for the subsequent layers, facilitating video frame reconstruction. The change in video frame size after upsampling in each NoRVC layer is determined by the upsampling parameters of each layer; specific values can be found in Section 4. Except for the first layer of NoRVC, the number of channels in each layer is half the size of the channels in the upper layer, with the number of channels set to 384 for the first layer in this paper. The output layer responsible for outputting RGB video images has a channel size of 3, and the output video size is the same as the original video size. For more details about the NRVC network, refer to Section 3.

The contributions of this paper can be summarized as follows:We propose a novel model of multiscale representations for video compression (MSRVC) for video compression, which contains multiple NoRVC blocks that effectively extends feature information to improve the quality of reconstructed images.We discover that inserting the FECA attention module in the MSRVC network can significantly improve the feature extraction performance of the model. This effectively enhances the network’s performance without significantly increasing its complexity.We demonstrate that NRVC has a 2.16% increase in decoded PSNR compared to the NeRV method at similar bits per pixel (BPP). Meanwhile, NRVC also outperforms the conventional HEVC [15] in terms of peak signal-to-noise ratio (PSNR).

## 2. Related Work

### 2.1. Traditional Deep Video Compression

As fundamental research in computer vision and image processing, video compression has been studied for decades. Prior to the emergence of deep neural networks, manually crafted image compression approaches such as JPEG [16] and JPEG2000 [17] were widely used. Many classical video compression algorithms, such as MPEG [18], AVC [19], HEVC [15], and VVC [20], have been proposed and gained broad influence. During the past decade, numerous visual compression approaches succeed in applying deep learning networks to optimize structures within the architecture of the traditional video compression pipeline. In the context of video compression based on deep learning, it is customary to apply a deep neural network to a specific module; for example, Li et al. [21] proposed a deep learning approach applied in the intra-frame coding of VVC, to predict the coding unit of the multi-type tree, ultimately resulting in a 50% reduction in the encoding time for VVC. Liu et al. [22] focused on the predictive coding of the feature space of low-dimensional frames in videos, developing a generative adversarial network for inter-frame prediction and convolutional long short-term memory network for future frame prediction. The compression ratio is better than AVC, HEVC, and other conventional video compression algorithms, even superior to or equivalent to previously proposed end-to-end depth video compression algorithms.

### 2.2. End-to-End Deep Video Compression

Since the limitations of deepening individual modules, an increasing number of researchers have been devoted to constructing end-to-end deep video compression schemes, where all modules are implemented based on deep neural networks. Lu et al. [7] proposed the first end-to-end deep model for video compression, which jointly optimizes all components of the video compression pipeline. The model utilizes two neural networks with an auto-encoder design to compress the corresponding motion and residual information, employing learning-based optical flow estimation to collect motion information and reconstruct the current frame. Li et al. [23] propose a deep video compression framework for conditional-coding compression using high-dimensional features in the feature domain context for encoding and decoding, building upon conditional coding structures to learn the temporal context of the current frame from propagated features. Sheng et al. [5] proposed a temporal context mining module with a hierarchical structure and reintegrated it into relevant modules of the compression model. However, these approaches still adopt the framework of conventional compression algorithms, and avoiding generating a large number of redundant parameters becomes a significant factor limiting the compression capability.

### 2.3. Implicit Neural Representation

Implicit neural representation is an innovative approach to parameterizing a variety of input information. Due to the powerful modeling plasticity of its treatment of representation information, INR has the capability of parameterizing the information represented through a neural network into a convergent function that maps a coordinate to corresponding values, for example, the pixel coordinates of an image and the RGB values of a pixel. Typically, INR approximates this function by the MLP combined with high-frequency sinusoidal activations [24] or Gaussian activations [25]; hence, the parameter characteristics of the input data are determined by the parameters of each layer of the network. INR is widely employed in various vision tasks, such as 2D videos [26,27], 3D shape reconstruction [28,29], 3D scene construction [30,31], and novel view synthesis [32]. In particular, COIN [33], which implements image compression with INR, demonstrates the feasibility of the application of INR in the video compression field. All above INR approaches are pixel-based implicit representations; when dealing with large samples and high-pixel datasets, the low efficiency of training and testing makes them inapplicable in several practical scenarios. By contrast, NeRV [8] uses image-based implicit representation and further combines convolution with INR for video image synthesis, thereby having the ability to reduce the expenditure of data processing and improving the efficiency of model training. However, the whole design of the model layers of NeRV is relatively rough, causing restrictions in extracting video frame features.

## 3. The Proposed Approach

### 3.1. Overview of Model Framework

The overall framework of NRVC is shown in Figure 1. As an image-based implicit representation method, the proposed NRVC can express any video input $V={\left\{v_{t}\right\}}_{t=1}^{T}\in R^{T\times H\times W\times 3}$ by the mapping function $f_{\phi }:R\to R^{H\times W\times 3}$. The mapping function is parameterized by the deep neural network ϕ, $v_{t}=f_{\phi }\left(t\right)$. The input of the mapping function is the value of the scalar frame index of standardized t∈[0,1], and the output is the corresponding RGB image $v_{t}\in R^{H\times W\times 3}$.

### 3.2. Positioning Encoding

Despite the fact that neural networks are universal function approximators, we have observed that training the entire network $f_{\phi }$ directly on timestamp *t* leads to a suboptimal performance of capturing high-frequency variations in color details and geometry details of the original video frames. This observation aligns with the conclusions of studies by [8,34] that deep networks tend to learn low-frequency functions. Furthermore, it has been verified that mapping the timestamp *t* to the high embedding space can obtain high-frequency data information and enhance the fitting effect of the neural network.

In previous studies [35], Fourier feature mapping has demonstrated its ability to assist MLP in learning frequency domain information in a range-controlled manner, allowing MLP to capture higher frequency content. In NRVC, we incorporate Fourier feature mapping as a positional embedding function to obtain a high-frequency representation of the timestamp input. By adjusting the parameters *b* and *l*, we can control the frequency of the timestamp input, thereby regulating the learning capacity of the MLP network. Formally, we utilize the following encoding function as the input embedding for NRVC:(1)$$\begin{array}{c} \rho \left(t\right)=(\sin\left(b^{0}\pi t\right),\cos\left(b^{0}\pi t\right),\ldots ,\sin\left(b^{l-1}\pi t\right),\cos\left(b^{l-1}\pi t\right)) \end{array}$$
where *b* and *l* are the hyperparameters of the network, *b* is used to set the va of input embedded data, and *l* is used to set the corresponding dimension of embedded data.

### 3.3. MSRVC

After extracting video sequence features from multi-frequency input in MLP layers, this feature information is fed to the MSRVC network. MSRVC contains FECA, five NoRVC blocks, and four upsampling blocks. The primary function of MSRVC is to reconstruct output video frames, which have the same size and comparable visual performance as the input frames.

#### 3.3.1. FECA Block

Due to the lack of sensitivity to local features in MLP, the extraction of local features is insufficient. Therefore, to further extract local features, we proposed the FECA attention mechanism as the first layer of the MSRVC. FECA avoids feature dimension reduction while capturing inter-channel interaction information, and divides the weights to improve feature extraction performance.

Our FECA block is presented in Figure 2. Given the aggregated feature *T* obtained from the upper MLP, FECA performs a fast one-dimensional convolution with a size of *n* to recompute channel weights. The value of *n* is adaptively determined based on the mapping of the channel dimension *C* that transmits the feature, representing the number of neighboring channels considered when capturing local cross-channel interaction information for each channel.

The size of *n*, which represents the extent of inter-channel interaction, is directly proportional to the channel dimension *C*. The input data dimension of the positional embedding increases together with the value of the positional embedding dimension parameter *l*. Consequently, more neighboring channels must be taken into account by each channel for interaction. This relationship implies that the positional embedding dimension parameter *l* is directly proportional to *n*. Furthermore, the channel dimension *C* is typically set as a power of 2, and *l* is usually set as a multiple of 10. Finally, The final calculation formula for the fast one-dimensional convolution is as follows:(2)$$n=\xi \left(C,l\right)={\left|\frac{\log_{2}^{C}}{\zeta }+\frac{l}{10\mu }\right|}_{odd}$$
where ${\left|a\right|}_{odd}$ indicates the nearest odd number of *a*. In our all experiments, we set ζ and μ to 2 and 1, respectively.

The weights *w* for regenerating each channel can be calculated through a fast one-dimensional convolution employing a kernel size of *n*:(3)$$w=\delta (ConvF1D_{n}\left(x\right))$$
where δ is the channel weight parameter, with the default set to 1. ConvF1D denotes the computation of one-dimensional fast convolution, and *x* represents the original channel weight parameters.

#### 3.3.2. NoRVC Block

The feature maps extracted by the feature extraction module are resized to the original video resolution through multiple NoRVC blocks. The structure of the proposed NoRVC module is illustrated in Figure 3. The training process of the NRVC network involves overfitting the video input data, with the NoRVC module being the main training network. We found that by sampling the shallow features and fusing them with the current features through a normalization network [36], we can accelerate the overfitting process and obtain higher-quality results. Therefore, the NoRVC is proposed to enhance the various level features of the current frame based on the upper layer module’s output. After being normalized, these features are fused with the current video frame features upsampled by PixelShuffle [37]. Finally, the combined output is activated through an activation layer.

In the NoRVC block, $F_{s}$ is the feature information of the current video frame after upsampling, and $F_{o}$ represents the feature input upsampled from the previous network layer, which can be obtained via multiple upsampling iterations from the output of FECA. Due to the different dimensions of $F_{s}$ and $F_{o}$, it is challenging to merge their features. Therefore, dimension synchronization is essential. By utilizing the upsampling layer to upsample $F_{o}$, we obtain $F_{u}$, which shares the same dimensions as $F_{s}$, carrying distinct feature information. After normalized $F_{u}$ is joined with $F_{s}$, the fusion feature $F_{n}$ is finally obtained, and the calculation process can be calculated by the following equation:(4)$$\begin{array}{c} F_{n}=\Phi \left(F_{s},F_{o}\right)=softmax\left(norm\left(F_{u}\right)\right)⊕F_{s}=softmax\left(norm\left(\epsilon \left(F_{o}\right)\right)\right)⊕F_{s} \end{array}$$
where Φ represents the mapping function corresponding to the feature fusion performed by $F_{o}$ and $F_{s}$, ϵ denotes the process of upsampling $F_{u}$, and norm represents the channel average variance normalization.

### 3.4. Loss Function

In the training of NRVC, we adopt the loss function of NeRV [8], which combines L1 [38] and SSIM [39] loss for network optimization, to calculate the loss of the predicted image and the baseline image at all pixel positions as follows:(5)$$\begin{array}{c} L=\frac{1}{N}\sum_{t=1}^{N}\left(\alpha \right\|f_{\theta }\left(t\right)-v_{t}\left\|+\left(1-\alpha \right)\left(1-SSIM\left(f_{\theta }\left(t\right),v_{t}\right)\right)\right) \end{array}$$
where *N* is the number of frames of the training video sequence, $f_{\theta }\left(t\right)$ is the prediction result of NRVC, $v_{t}$ is the baseline truth value of the corresponding frame for each timestamp, and α is a hyperparameter that balances the weight of each loss component.

## 4. Experiments

### 4.1. Datasets and Settings

Scikit-video is a Python library for processing video data that provides four sample datasets, all with a resolution of 1280 × 720. Among them, the “Big Buck Bunny” sequence, with a total duration of 5 s, is used for general video processing tasks. The “Bikes” sequence, with a total duration of 10 s, is used for scene detection tasks. The “Carphone” pristine and distorted sequences, each with a total duration of 4 s, are used for full-reference quality algorithm testing.

The Ultra Video Group (UVG) [40] dataset consists of 16 multi-purpose 4K (resolution of 3840 × 2160) test video sequences, including close-ups of people and animals, horse racing, and natural landscapes. These videos were captured using a 16-bit F65RAW-HFR format on a Sony F65 camera and were converted to 10-bit and 8-bit 4:2:0 YUV videos using FFmpeg, a comprehensive, cross-platform solution for recording, converting, and streaming audio and video. Of the 16 videos, 8 were captured at a speed of 120 frames per second (FPS), whereas the rest were captured at a speed of 50 FPS, resulting in a total of 9000 frames in the dataset.

We performed quantitative and qualitative experiments on eight different video sequences collected from scikit-video and UVG datasets to compare our NRVC with other video compression approaches. The video sequence of scikit-video is the “Big Buck Bunny” video sequence with a resolution of 1280 × 720 and a total number of 132 frames, and the UVG dataset has 7 videos with a resolution of 1920 × 1080 and 200 frames per video, for a total of 1400 frames.

### 4.2. Implementation Details

We use the Adam [41] optimizer to train the entire network with the learning rate set to $5e^{-4}$. During training, we use a cosine annealing learning rate table [42] and set the warmup epochs to 20% of all epochs. There are five NoRVC blocks in our whole NRVC, and the upsampling factors are adjusted according to the resolution of the video sequence. In the following experiments, the upsampling factors are set to 5, 2, 2, 2, and 2, respectively, for the “Big Buck Bunny” video sequence, and 5, 3, 2, 2, and 2, respectively, for the UVG video sequences. For the position embeddings entered in (Equation 1), we set b=1.25 and l=80 as our default settings. For the target loss function in (Equation 5), α is set to 0.7 by default. We evaluate the video quality with two metrics: PSNR and MS-SSIM [43].

Once completing the video fitting, then the model compression is used to accomplish the purpose of video compression. For fairness of the experiment, we perform video compression with the same method as NeRV. The model compression process consists of three steps: model pruning, weight quantization, and weight encoding. We achieve model size reduction by global unstructured pruning. In our experiments, model quantization is performed after model training.

We built the model using PyTorch and trained it at full precision (FP32). All experiments are based on NVIDIA RTX3090.

### 4.3. Comparative Results

Firstly, we compare our method with NeRV [8]. The training datasets include the “Big Buck Bunny” video sequence and the UVG video sequences. Due to equipment limitations, we used the NeRV model with moderate parameters for the control group experiment. Therefore, we constructed NRVC, a network with a similar scale as NeRV, by resizing the width of the convolutional filters for comparison. We use PSNR, MS-SSIM, and decoding FPS as indicators to evaluate the quality of the reconstructed video. Table 1 shows the results of the comparison, in which NRVC and NeRV were trained for 1200 epochs. Compared with the NeRV method, NRVC improves the decoded image quality. As a result of the residual network’s utilization, the number of model parameters rises and the decoding speed falls slightly, but stays within NeRV’s range of magnitude. We also conducted multiple trials with different training epochs, and the results are presented in Table 2, which demonstrates that longer training times can lead to significantly superior overfitting results. Compared to NeRV, NRVC has superior reconstructed video quality under the same training settings. Table 3 displays the performance of NRVC and NeRV in reconstructing videos under different training epochs for the “Big Buck Bunny” dataset and the UVG dataset. For the UVG dataset, the training batch size *b* is set to 7. NRVC still demonstrated superior video reconstruction capabilities to NeRV.

### 4.4. Video Compression

Then, we compare the compression performance of NRVC with other compression approaches on the UVG dataset. The results for different pruning ratios in model pruning are shown in Figure 4, where the model with 40% sparsity still obtains performance that is equivalent to the complete model. The results of model quantization are shown in Figure 5, where the model with 8 quantization bits still maintains the reconstructed video quality in comparison to the original model with 32 bits.

Figure 6 and Figure 7 show the curves where we compare our proposed approach with HEVC [15], NeRV [8], and HLVC [44] on the UVG dataset. HEVC is executed in medium preset mode. The HLVC is an end-to-end hierarchical video compression method that improves compression efficiency by dividing frames into quality levels and recursively associating frames with neighboring high-quality frames. However, In Figure 6, as the BPP decreases and overall frame quality declines, the proportion of high-quality frames also rapidly decreases. Consequently, at low BPP, its PSNR values are lower than those of NRVC and NeRV. Our approach NRVC basically outperforms the NeRV method.

Figure 8 shows the visualization results of the decoded video frames, where NRVC presents a better capacity for reconstructing more visual details with higher quality at similar compression conditions. In Figure 8, the PSNR of NeRV is 34.21 and the PSNR of NRVC is 34.95; NRVC has a 2.16% increase in decoded PSNR compared to the NeRV method.

### 4.5. Ablation Studies

Finally, we conducted ablation investigations on the “Big Buck Bunny” dataset, where PSNR and MS-SSIM are utilized to evaluate the rebuilt video. The ablation comparison experiments of NeRV [8], NeRV only with FECA, NeRV only with NoRVC, and NRVC are implemented. Table 4 shows the reconstructed video results of different models in ablation experiments. Figure 9 shows the visible results of the reconstructed video with different approaches. As can be observed, the model containing FECA or NoRVC shows more visual performance than the original model. Moreover, NRVC displays richer and more accurate image details than other models.

### 4.6. Resolution Studies

NRVC can maintain the same resolution of the reconstructed video frames as the input video by adjusting the dimension parameters of the MLP’s final output layer and the upsampling parameters of each NoRVC layer. In this section, we select four video sequences from the UVG dataset and modify the video frame resolutions using the ffmpeg tool. Subsequently, NeRV was trained on all frames of these videos with different resolutions to evaluate the compression capability for videos with varying resolutions of NRVC. As shown in Figure 10, we conducted experiments on the “ShakeNDry”, “YachtRide”, “Bosphorus”, and “HoneyBee” datasets to assess NRVC’s compression capability for videos with resolutions of 800 × 720, 1280 × 1024, 1600 × 1200, and 2128 × 2016, respectively. NRVC exhibits good compression and reconstruction capabilities for video frames of any resolution. Despite having a smaller range of video resolutions that it can encode compared to traditional encoding methods, NRVC is still capable of compressing videos of the majority of resolutions.

## 5. Conclusions

In this paper, we explore a novel neural implicit representation for video compression. Compared to end-to-end deep network video compression, we discover that INR-based compression methods demonstrate faster encoding and decoding speeds. Meanwhile, by expanding the feature information during feature fusion, we can accelerate the network’s fitting speed. At high compression rates, reconstructed video frames show improved image performance. In addition, approaches with superior compression performance than conventional video compression representations will be presented based on this method in the future, taking into account the enormous potential of neural implicit methods.

## Figures and Tables

**Figure 1 entropy-25-01167-f001:**
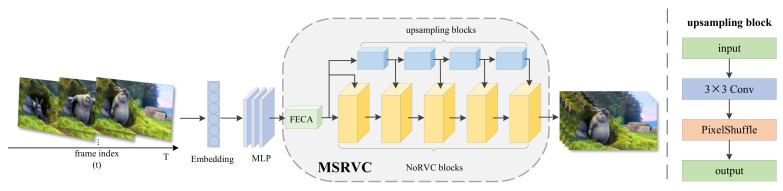
The framework of NRVC consists of three components: position encoding, MLP, and MSRVC. MSRVC includes FECA, five NoRVC blocks, and four upsampling blocks. Each upsampling block is composed of a 3 × 3 Conv and a PixelShuffle layer.

**Figure 2 entropy-25-01167-f002:**
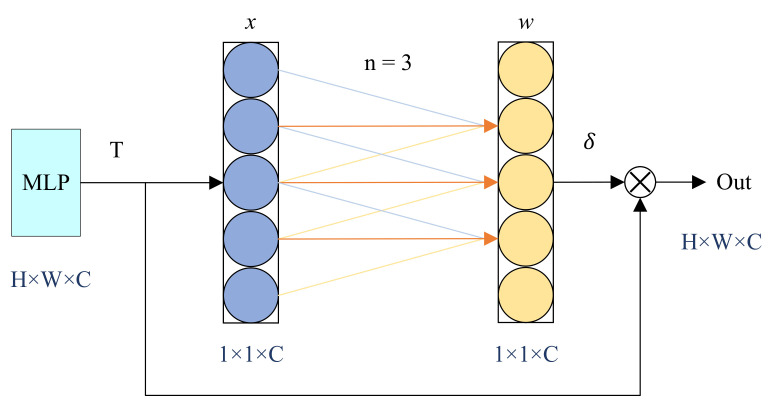
FECA block, where *x* represents the original channel weight parameters and *w* represents the regenerated ones.

**Figure 3 entropy-25-01167-f003:**
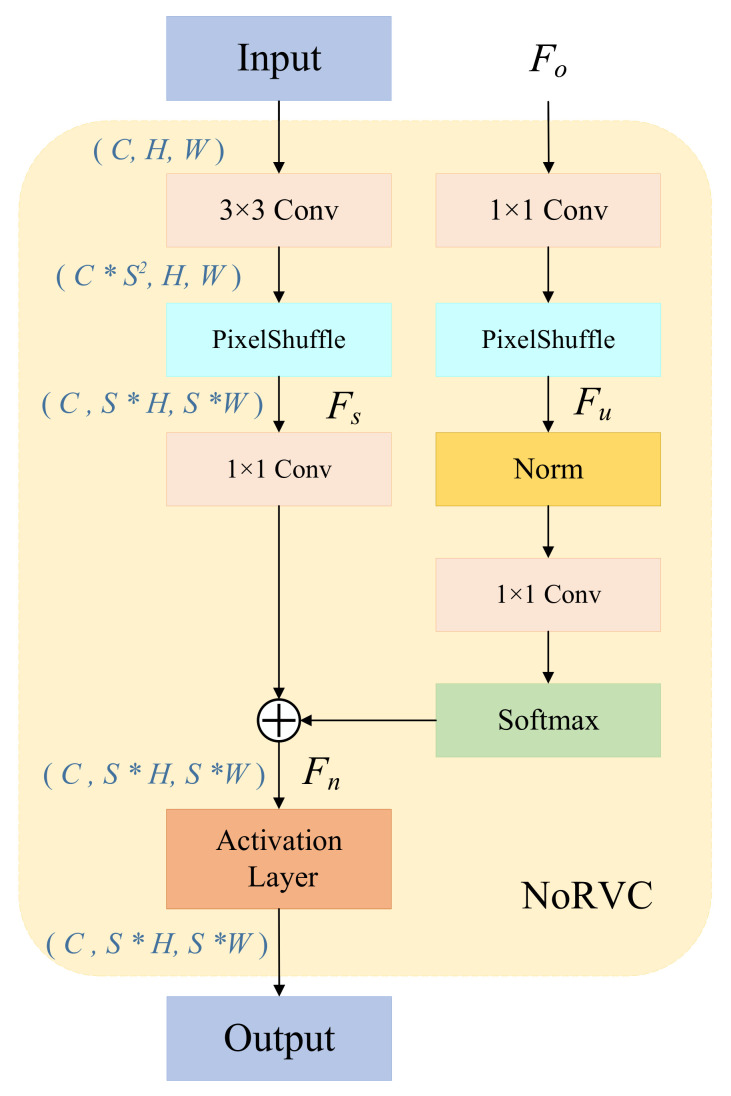
NoRVC block, where S is the upsampling factor.

**Figure 4 entropy-25-01167-f004:**
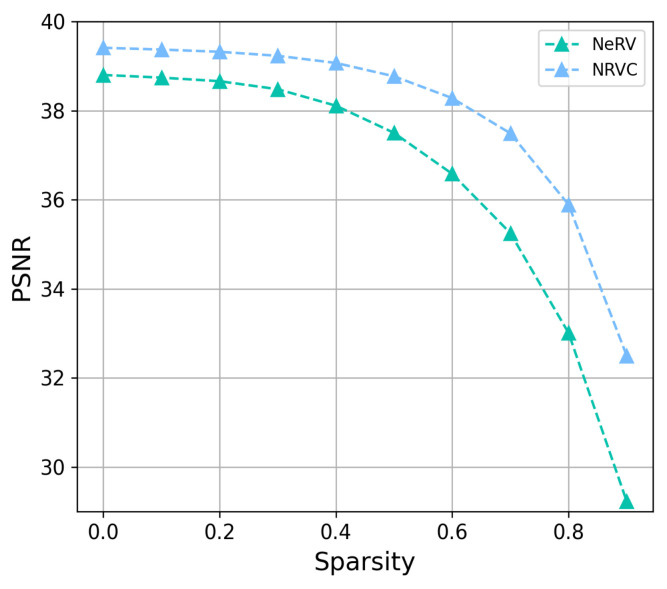
Comparison results of model pruning. Sparsity is the ratio of parameters pruned.

**Figure 5 entropy-25-01167-f005:**
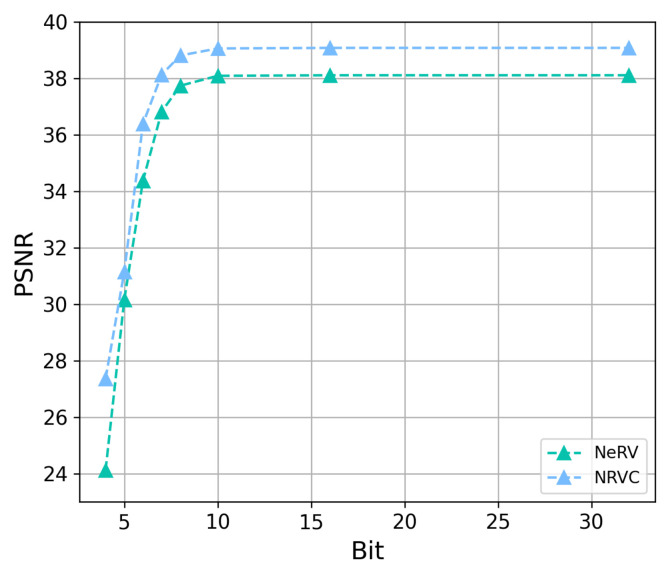
Comparison results of model quantization. The bit refers to the number of bits used to represent the parameter value.

**Figure 6 entropy-25-01167-f006:**
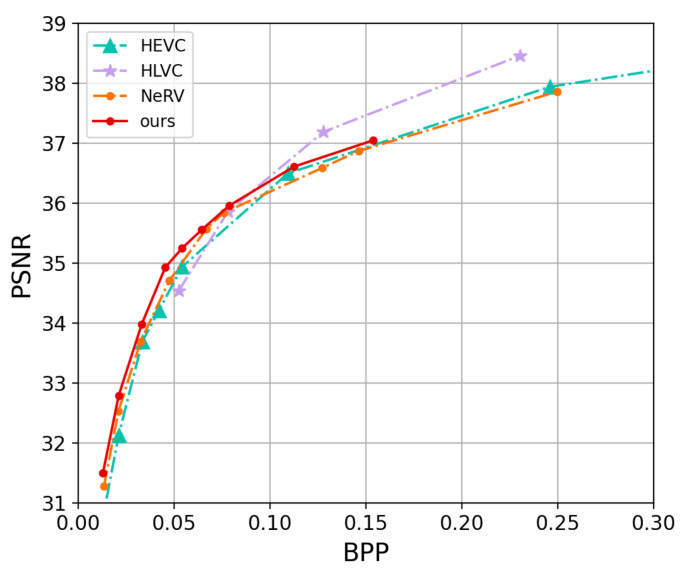
PSNR vs. BPP on UVG dataset.

**Figure 7 entropy-25-01167-f007:**
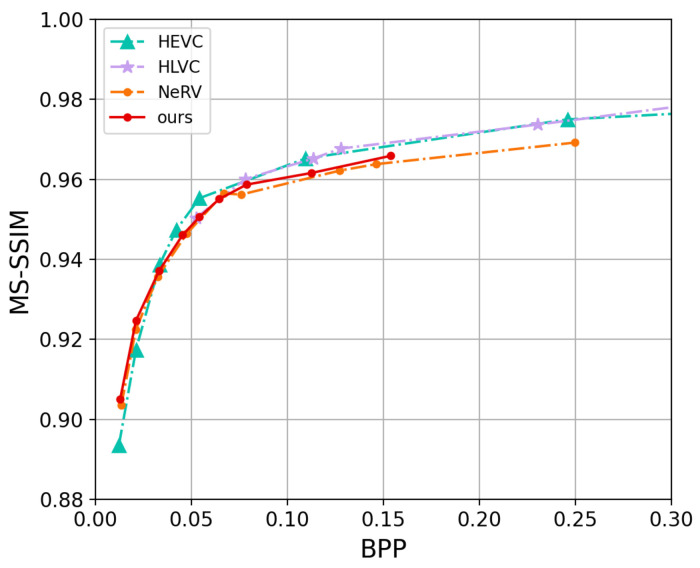
MS-SSIM vs. BPP on UVG dataset.

**Figure 8 entropy-25-01167-f008:**
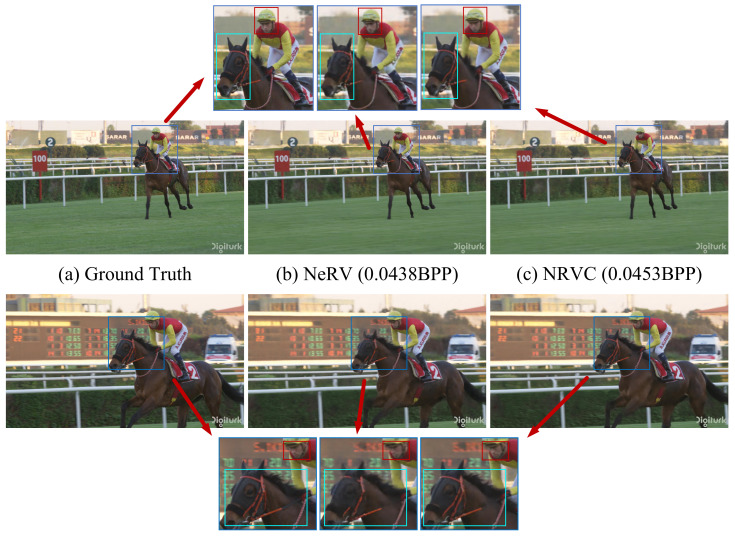
Video compression visualization. (**a**) is the original image. (**b**,**c**) are, respectively, the decoded frame of NeRV and NRVC on the UVG dataset. The red and light blue boxes show more specific details in each group of images. Obviously, (**c**) shows better details in the reconstruction of the rider’s face and horse than (**b**).

**Figure 9 entropy-25-01167-f009:**
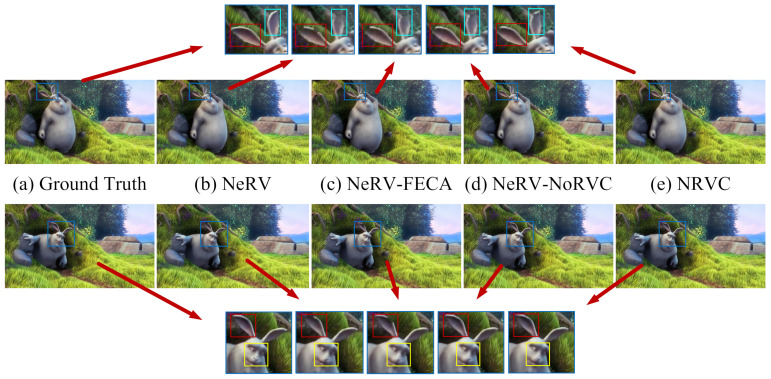
The visual results of ablation studies. (**a**) is the original image. The red and light blue boxes show more specific details in each group of images. As the final model, (**e**) shows more image details than (**b**–**d**) in the reconstruction of rabbit ears.

**Figure 10 entropy-25-01167-f010:**
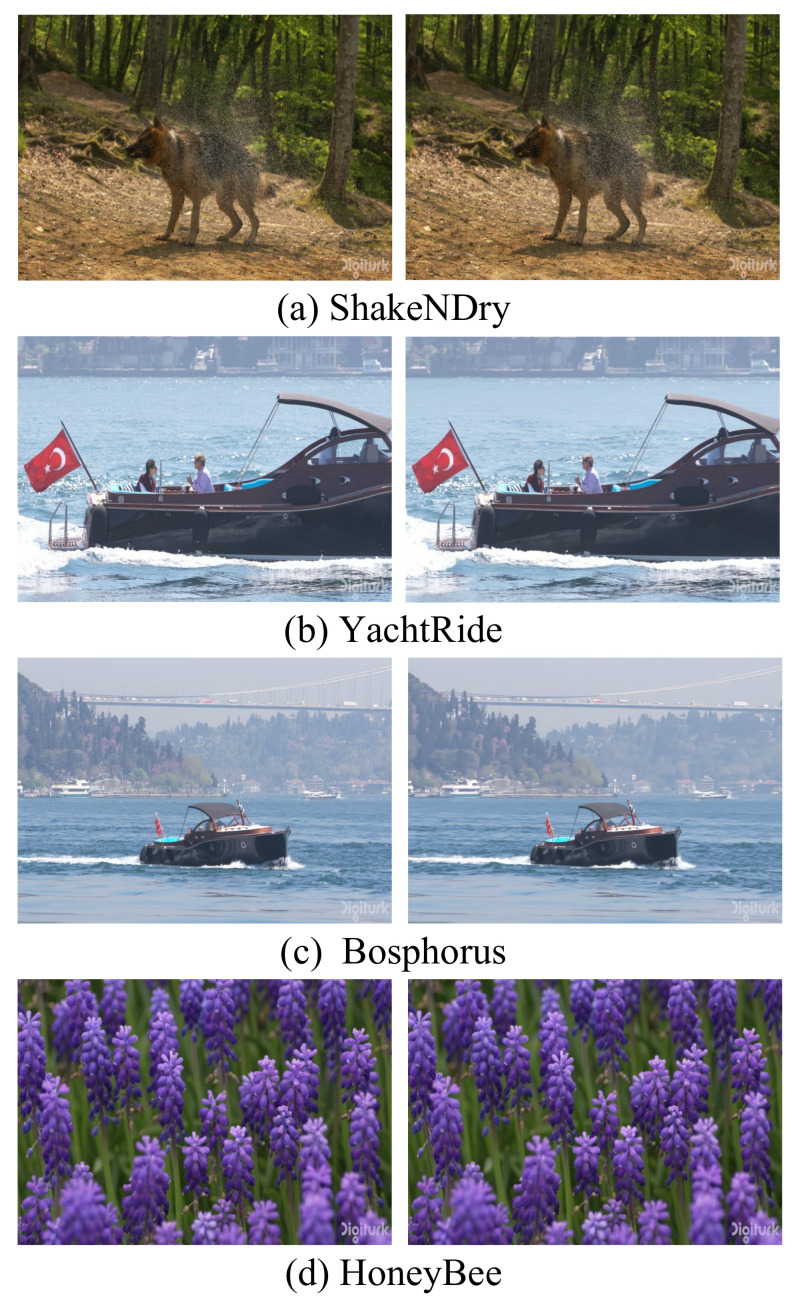
Different resolution research. In (**a**–**d**), the left is the original video frame and the right is the reconstructed frame with NRVC. In (**a**), we use the “ShakeNDry” dataset with 800 × 720 resolution. In (**b**), we use the “YachtRide” dataset with 1280 × 1024 resolution. In (**c**), we use the “Bosphorus” dataset with 1600 × 1200 resolution. In (**d**), we use the “HoneyBee” dataset with 2128 × 2016.

**Table 1 entropy-25-01167-t001:** NRVC and NeRV [8] were trained for 1200 epochs. The number of channels after position encoding is set to 8.

Methods	Parameters	PSNR	Decoding Fps	MS-SSIM
NeRV	5.0M	38.50	**34.98**	0.9910
NRVC	5.2M	**39.19**	31.60	**0.9921**

**Table 2 entropy-25-01167-t002:** Comparison between NRVC and NeRV on the “Big Buck Bunny” dataset with different epochs. The number of channels after position encoding is default set to 8.

Epoch	NeRV	NRVC
300	33.53	**34.10**
600	36.43	**37.00**
900	37.78	**38.35**
1200	38.50	**39.19**

**Table 3 entropy-25-01167-t003:** The results of NRVC and NeRV in comparison between different datasets.

Datasets	Epoch	NeRV	NRVC
bunny	600	36.43	**37.00**
bunny	1200	38.50	**39.19**
UVG	700	30.97	**31.47**
UVG	1050	31.58	**32.18**

**Table 4 entropy-25-01167-t004:** The results of ablation studies, including NeRV, NeRV only with FECA, NeRV only with NoRVC, and NRVC. NRVC shows richer and more accurate image details.

Methods	PSNR	MS-SSIM
NeRV	38.50	0.9910
NeRV-FECA	38.69	0.9914
NeRV-NoRVC	38.87	0.9917
NRVC	**39.11**	**0.9921**

## Data Availability

Not applicable.

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
