# Peer review of "NRVC: Neural Representation for Video Compression with Implicit Multiscale Fusion Network"

_entropy, 2023, doi:10.3390/e25081167_

Round 1

Reviewer 1 Report

1. Many abbreviations are used in this paper. However, some of them are shown without listing their complete forms first, such as NeRV and FECA, etc. Although the complete forms appear in the abstract, we think that they have to be listed in the main text.

 2. The title “NRVC: Neural Representation for Video Compression” is quite general and looks like a review paper. However, a specific scheme is proposed.

 3. Sec. 3 is the key of this paper but it is just too short. Eq(1) is shown without explaining why cosine/sine functions are used. Sec. 3.3.1 contains only some descriptions but it is not easy for readers to appreciate the novelty.

 4. Compared with traditional video coding, the proposed method demonstrates better performance. However, complexity could be a problem, which may affect the feasibility of deep video coding. Besides, using such methodology may only work in videos with a fixed resolution. Nevertheless, traditional video coding may not have such limitations.

 5. It seems that showing the video “Big Buck Bunny” in the experimental results may hint that the proposed scheme may only work on such animation videos. Some examples of natural videos can be shown in this section. Besides, the videos are all very short. We are not sure whether the experiments in such short videos are convincing.

Presentation is OK. Too many abbreviations are used. 

Reviewer 2 Report

A new NRVC network has been developed for video compression. Sufficient experimental studies have been made by comparing with other state-of-the-art methodology. The comparison results verify the effectiveness of the proposed approach.

Author Response

Thank you for your Comments. 

Round 2

Reviewer 1 Report

1. The difference between NERV and NeRV should be addressed. We are unsure if the basic structure is similar (although the authors briefly mentioned the drawback of NeRV) as they are compared in this 

2. The authors explained the proposed scheme using Fig. 1. However, it is not easy for people with little experience in such video coding practice to know only the operations from this figure.

3. Since HLVC, which shows very high PSNR, is mentioned and compared, the authors may add some comments about HLVC.
